# Human Monkeypox—A Global Public Health Emergency

**DOI:** 10.3390/ijerph192416781

**Published:** 2022-12-14

**Authors:** Enrico Maria Zardi, Camilla Chello

**Affiliations:** 1Internistic Ultrasound Service, Department of Medicine and Surgery, Fondazione Policlinico Universitario Campus Bio-Medico, Via Alvaro del Portillo, 200, 00128 Rome, Italy; 2PhD Course, Department of Medicine and Surgery, Fondazione Policlinico Universitario Campus Bio-Medico di Roma, 00128 Rome, Italy

**Keywords:** COVID-19, epidemic, human monkeypox, therapy, transmission, vaccinia, virus, zoonosis

## Abstract

Monkeypox, a viral zoonosis caused by an Orthopoxvirus, is clinically characterized by fever, headache, lymphadenopathy, myalgia, rash and burdened by some complications that can be severe and life threatening. Monkeypox, endemic in some central and west African countries, in tropical areas near equator, rose to the headlines following its recent outbreak in non-endemic countries of Europe and the USA. Thus, the World Health Organization, worried about the growing dimension of the problem, declared monkeypox a global public health emergency. Now, after months of careful observation, the western scientific research is drawing conclusion that African endemic countries represent a reserve pool able to feed, through travelers and sexual networks, the outbreak in non-endemic countries in which high-risk communities such as gay and bisexual men are the most affected. Prevention through vaccination and early diagnosis are the core to breaking the chain of diffusion of this epidemic. Particular attention should be paid to avoid the spread from endemic countries, also implementing the economic investments in their public health system. Information campaigns and assistance to high-risk classes in non-endemic countries are important priorities, however, assuming that specific treatments for this disease are still tentative.

## 1. Introduction

Monkeypox, a double-stranded DNA virus (genome size about 200 kilobases, seven times larger than SARS-CoV-2), has been known to belong to the Orthopoxvirus genus of the Poxviridae family since 1958 when, after two outbreaks of a generalized vesicular-pustular rash illness in colonies of Cynomolgus monkeys, it was identified at the Statens Serum Institute of Copenhagen [1,2,3] (hence the name monkey pox, although rodents would seem to be the main viral reservoir). In the years following the discovery, some outbreaks of pox infection were recognized in the United States of America (USA), Denmark and the Netherlands among groups of captive monkeys (of some species) (Cynomolgus, Rhesus, Macaca philippensis, Macaca mulatta, Macaca irus, Langur) coming from some regions of Asia (Singapore, India, Philippines, Malaysia) [4].

Monkeypox virus is not ancestral to variola virus (the causative agent of smallpox), but both have evolved independently of their common ancestor and the possibility of one becoming transformed into the other has been excluded, although they are closely related [5,6].

The presence of human monkeypox was first ascertained in September 1970 in a 9-month-old child in Basankusu district, on Equateur Region, in the Republic Democratic of the Congo [7]. The outcome of the disease was complicated by the onset of other infections leading up to child’s death [7]. Since then, some outbreaks of human monkeypox, a few of which also causing death, have been identified in more than 50 years in several regions of the world. The incidence of monkeypox has risen almost tenfold, over the past few years, going from 0.63 to 5.53 in 10,000 people [8,9].

Clinically, human monkeypox has similar but milder features than those of smallpox; surely it has a different epidemiology [10]. Initially, two clades of monkeypox virus strains, with about 0.5% genomic sequence difference, were recognized in some regions of Africa, one of which (Congo basin MPXV-ZAI-V79) is more virulent than the other (West Africa) [11]. Now, in accordance with the World Health Organization (WHO), a novel classification of monkeypox virus strains was proposed based on three clades (clade I corresponding to Congo basin and clades IIa and IIb to West Africa) [12].

Although the WHO in 2018 still considered the monkeypox a rare viral zoonotic disease of remote areas of central and West Africa, near tropical rainforests [13], on 23 July 2022 deemed it a public health emergency of international concern, for the moment concentrated among men who have sex with men, especially those with multiple sexual partners, giving few necessary justifications to substantiate its decision [14,15]:-the rapid spread of the virus to many countries never seen before and the new modes of transmission, largely unknown, meeting the International Health Regulations criteria.-the risk for human health and its potential implication with the international traffic;-the high risk in the European regions and the moderate risk worldwide.

In the light of this, to avoid misinformation and unjustified public concern [16] this review aims to deepen understanding but also to summarize the latest findings on this infection, particularly with regard to epidemiology, prevention, clinical features, diagnosis, management and strategies adopted to deal with it.

## 2. Epidemiology

Monkeypox is a rare infectious zoonotic disease which is endemic in central and western Africa areas (especially in Democratic Republic of the Congo, Liberia, Sierra Leone, Nigeria and Ivory Coast) that are characterized by high mean annual precipitations and low altitude [17,18]. From its discovery to 2000, other regions in Benin, Cameroon, the Central African Republic, Gabon, Ghana, Sierra Leone and South Sudan were added to the list of the endemic areas for monkeypox, in Africa. However, the highest number of cases of monkeypox was reported in the Democratic Republic of the Congo with a case fatality rate of 11%, in smallpox unvaccinated patients, that reached 15% among children under 4 years [17].

In the Democratic Republic of the Congo in 1996–1997 and in 2003, the first outbreaks of human monkeypox have occurred in which it was observed that transmission of monkeypox virus took place from person to person [19,20].

In particular, during the 1996–1997 epidemic, the interhuman mode of transmission continued for two years in large areas of the Katako-Kombe and Lodja health-care zones [19].

On May 2003 there was the first documented human monkeypox case outside of Africa, in a 3-year-old girl from central Wisconsin in the USA [21]. An interesting epidemiological study showed that the monkeypox virus was present in African rodents arrived in a shipment from Ghana and responsible for transmitting the virus to animals destined for the pet trade [22]. In the human American outbreak that followed, involving six states (Illinois, Indiana, Kansas, Missouri, Ohio, and Wisconsin), fatal cases were not reported [22], probably because the West African virus, the unique viral strain isolated in this outbreak, was less virulent than the other strain prevalent in the Congo [11,23]. Furthermore, in these states of the USA, most patients had performed vaccination against smallpox in the past [22].

Interestingly, it was noted that the outbreak occurred in the USA was strictly related to the domestic use of exotic wildlife which pushed the Food and Drug Administration to ban the import of all rodents from Africa and the sale, distribution, transport, and release into the environment of prairie dogs and six African rodent species [22]. However, the spread of monkeypox virus through human contact during this outbreak could not be excluded although the exposure to infected prairie dogs was the main invoked mechanism of transmission to humans [21].

The analysis of another outbreak of monkeypox that began in South Sudan in 2005, confirmed the role of importation into the area from a place where monkeypox is endemic (the Congo) both through infected animals and humans entering in that area, thus supporting the hypothesis of person-to-person transmission [24].

However, a different epidemiology distinguished the African from the American outbreak.

Indeed, in Africa, human monkeypox primarily affected children younger than 10 years (80% of cases) that were part of rural populations (residents of villages of less than 1000 people), and the transmission was both via contact with infected small mammals obtained for food or through infected animal bites and person-to-person contact (28% of cases) with a case fatality ranging from 10% to 17% [25]. In the USA, an equal number of males and females was affected, infected prairie dogs were the main vehicle of transmission, whereas person-to-person transmission was infrequent and no death was observed [24].

Some conditions make possible the epidemic outbreak of monkeypox [26,27,28,29,30,31]:today, 70% of the world population and 75% of young individual have never received the smallpox vaccine resulting unimmunized against the monkeypox virus;re-emergence of monkeypox in Africa (epidemic in Nigeria 2017–2018 and in Central African Republic in 2018) due to endemicity favored by climate change, rain forest exploitation, geopolitical and armed conflicts, without neglecting the possibility of intrafamilial infections;travel-associated monkeypox (case worldwide).

It has become clear that Monkeypox may spread from human to human through close contact with skin lesions or lesions on internal mucosal surfaces, body fluids and respiratory droplets, but cases of airborne transmission were not reported [32,33]. An indirect transmission through contaminated objects is also mentioned [32].

Recently, in the course of the monkeypox epidemic in Nigeria in 2017–2018, the sexual transmission in several patients with genital ulcers was also hypothesized [34].

Travel associated monkeypox was reported in 2018 in men who returned from Nigeria to Israel, Singapore and United Kingdom [35,36,37,38,39] with the peculiarity, in the latter country, of the first nosocomial and household transmissions to be reported outside of the African continent [37].

They were also the first-time international travelers implicated in the spread of monkeypox outside of an outbreak setting [40].

Coming to the present day, the Democratic Republic of the Congo would be the greatest responsible for epidemic resurgence, with 1356 cases of monkeypox and 64 deaths from January to July 2022 [41,42].

In the first months of the current year, the sudden appearance of the monkeypox virus in non-endemic areas and the high risk of human-to-human transmission attracted the attention of the scientific and institutional world. This led to increase the amount of research and number of publications on this matter.

From January 2022 to June, a great number of cases of monkeypox have been reported also in several non-endemic countries of Europe, in the United Kingdom, USA, ASIA (Singapore, South Korea) Australia and Republic of China (Taiwan) (Table 1).

Important research across 16 countries showed that gay or bisexual men represented the 98% of the persons affected by monkeypox, 75% of whom were White, and 41% with human immunodeficiency virus infection [43].

Sexual transmission was re-proposed as an important way of infection precisely because the lesions were mostly located in genital and perianal sites, most patients were men who had had sex with men or multiple sexual partners who travelled to festivals or did not use the condom [44]. The common occurrence of a positive seminal fluid for monkeypox in men who had sex with men [45] provided further support for this.

Furthermore, even in the USA the transmission of monkeypox was prevalent in communities of gay, bisexual, and other men who have sex with men and in racial and ethnic minority groups, as well [46].

Physical contact, sexual transmission in men who have sex with men, or transmission via fomites were the infection routes indicated in a prospective cross-sectional study performed by experienced dermatologists in multiple medical facilities during the monkeypox epidemic in Spain in the 2022 outbreak [47].

Person-to-person transmission of monkeypox within the United Kingdom was also demonstrated in a retrospective observational analysis of patients between May and July 2022 in a south London High Consequence Infectious Diseases Centre [48].

Even in Germany, France and Portugal the analysis of the first monkeypox outbreak showed a clear human-to-human transmission among a susceptible demographic group (especially men who have sex with men) [49,50,51]. A probable importation of the virus in Portugal from other European countries, in which it circulated without being detected, was suggested [49].

The absence of connection between the current monkeypox outbreak and travels in the endemic areas strengthens the impression of an undetected spread of monkeypox virus in Europe through human-to-human transmission [52].

Indeed, the hypothesis of a mutation in the monkeypox virus able to explain the recent outbreak worldwide has been denied by the absence of major mutations in the viral genome [53].

Today, it is believed that the community transmission of monkeypox is becoming the main cause of contracting the disease rather than travels to monkeypox-endemic countries [54,55].

An interesting study evaluating the disease transmissibility for the current monkeypox epidemic within populations of men who have sex with men, detected a high reproduction number (R_0_) ranging from 1.4 to 1.8, but less than 2.13 observed in the monkeypox epidemic in the Democratic Republic of the Congo in 1980–1984 [56]. However, since R_0_ is higher than 1, in high-risk subjects, an expansion of epidemic is expected [57].

A meta-analysis of several research articles worldwide in a time span from 1980 to 2022, whose purpose was to assess the rate of hospitalization, the presence of complications and deaths from human monkeypox, concluded that 35% of patients (95% CI 14–59%) was hospitalized and 4% (95% CI 1–9%) had fatal outcomes [58]. No fatal outcome was observed during the monkeypox outbreak of 2003 in the USA, but 26% of patients needed to be hospitalized for more than two days [59].

On the contrary during the current monkeypox epidemic, some studies showed lower percentages of hospitalization than previous reports and no deaths: 6% in France [50], 8.3% in Germany [49] 8.8% in Milan (Italy) [60], 9.2% in the UK, [61], and 11.1% in Portugal [51]. **An interesting report highlighted that in the USA most of hospitalized patients with monkeypox were black men with HIV infection of which 21% died despite oral or intravenous tecovirimat administration, vaccinia immune globulin intravenous, intravenous cidofovir and intensive care unit-level care [62].**

## 3. Prevention

Various practical preventive measures should be taken and, among them, interrupting the virus transmission from animals to humans might greatly hamper the monkeypox epidemic from spreading [63]. For example, pets belonging to individuals with monkeypox should be kept at home and away from other animals and people up to 21 days after the most recent contact [63].

However, since as previously said, the human-to-human transmission is the prevalent route during this epidemic, it is essential to avoid [64]:close physical contact (including sexual activity) with persons affected by monkeypox rash;contact with body fluids of infected animals;bites of infected animals;processed meat from infected animals;become in touch with fomites.

Contact tracing is a desirable measure to identify exposed persons, isolate them and prevent further cases. However, it should be remembered that previous smallpox vaccination may give false positive results [65].

Since men who had sex with men, bisexual and also ethnic and racial minority groups are affected by monkeypox, the efforts of public health should, rather, be focused on these groups for infection prevention and testing [66].

Furthermore, it would be important to give medical staff any information such as to pay close attention to hand hygiene, ensure a proper cleaning of contaminated surfaces and, in addition, provide them with protective devices for full safety of health (for example, eye protection in the event of procedures that expose to the risk of contact with body fluid) [63,67].

## 4. Proposed Vaccines

Due to the cessation of smallpox vaccination for people born after 1980 and the poor immunity to monkeypox in younger individuals who live in non-endemic countries, vaccination program is of special importance in this context to reduce spread and severity of monkeypox disease and break the chain of transmission [68]. **News on vaccination coverage in Europe are lacking but it is known that there is no substantial difference between European countries in relation to the degree of complete vaccination against smallpox except for UK that has a low coverage** [69]**. There are, however, significant differences between zones within the same country which depend on some reasons such as demographic changes and different past vaccination strategies against smallpox** [69]**.**

According to some guidelines, people at risk of monkeypox, especially healthcare workers, should be given vaccinia [70].

The first next-generation smallpox vaccine is Dryvax vaccine administered via skin scarification. It was discontinued as it causes severe side effects such as Stevens-Johnson syndrome, acute vaccinia syndrome (characterized by malaise, fever, myalgia, and headache) and myocarditis [71].

The second next-generation vaccine is ACAM2000, a single-dose vaccine delivered via skin scarification, requiring four weeks after vaccination to give immune protection [72]. Unfortunately, also this vaccine was burdened with many side effects such as myocarditis, pericarditis, lymphadenitis, acute vaccinia syndrome. Pregnancy was contraindicated for 4 weeks after its administration [72].

The third next-generation smallpox vaccine is “MVA-BN (JYNNEOS in the U.S.. **Brand name: Imvamune and Imvanex in the European Union and Canada, approved from the European Medicines Agency on 22 July 2022**)”, a live but attenuated non-replicating MVA strain [73,74]. It may be administered both by subcutaneous route, with an injection volume of 0.5 mL twice 4 weeks apart (requiring 14 days after the second dose to reach a full immune protection) and intradermally, at a lower dose but with the same immunogenic response as the subcutaneous route in preventing monkeypox infection and illness [71,72,73]. This vaccine shows less serious side effects (headache, myalgias, lymphoadenopathy) than previous vaccines and is licensed for prevention of both smallpox and monkeypox [73].

Another potential third-next-generation smallpox vaccine, that is currently under investigation, is LC16m8. It is a replicating attenuated vaccinia strain, developed in Japan, able to diminish the production of the extracellular envelope of the monkeypox virus [75]. Given subcutaneously in non-human primates, it induced long lasting protective immunity against monkeypox virus, suggesting that it might be a suitable choice to protect people from human monkeypox in endemic regions [75].

Interestingly, the approved vaccines are now also recommended for post-exposure prophylaxis especially in children and pregnant women [70]. **Thanks to recent molecular docking studies, molecular dynamic and c-immune simulations, important progress is being made in research on vaccines against monkeypox with the hope that soon there will be interesting solutions to face this disease even better** [76]**.**

## 5. Pathophysiology and Clinical Features

The pathogenesis of human monkeypox includes an incubation period ranging from 5 to 21 days where some steps occur. **First, binding to glycosaminoglycans to enter the host cells.** Next, the viral replication at the site of inoculation localized in cytoplasm of mononuclear phagocytic cells (monocytes, macrophages, dendritic cells) epithelial cells and fibroblasts, which follows the spread of extracellular enveloped virions to lymph nodes and finally the viremia that favors the viral spread to other organ systems and, once again, the skin cells [68,77,78]. **During this period, the virus produces two infectious forms: extracellular enveloped virions and intracellular mature virions. A good protection against the infection depends on the ability of vaccines and antibodies to target these antigens [79].**

Interestingly, it was hypothesized that this virus can gain access to the central nervous system both via the olfactory epithelium and infected circulatory monocytes/macrophages [80].

Bearing in mind that the clinical presentation of vaccinated and unvaccinated individuals is similar [81], flu-like symptoms (fever, lethargy, headache, backache, sore throat) can appear for both groups from 1 to 3 days before the onset of the rash (Figure 1) [56]. According to CDC (Centers for Disease Control and Prevention) the most common signs and symptoms are: rash (98.6%), malaise (tiredness) (72.7%), fever (temperature above 38°C (72.1%), severe chills (68.9%), headache (65.2%), enlarged lymph nodes (mainly of the neck, groin, and submandibular region) (64.3%), myalgia (61.8%), and pruritus (60.2%) [82].

The rash develops on the face and even oral mucosa and then spreads centrifugally to the trunk and extremities having the potential to involve other zones such as the genitalia [67]. The lesions are generally 2–5 mm in diameter, but they can reach 10 mm and appear firm, deep seated, well circumscribed, painful, itchy, sometimes umbilicated [83].

The rash starts with a macular shape which subsequently becomes papular, vesicular, and pustular before crusts are formed [71,84].

The time period from rash onset to desquamation is between 14–21 days [81]. In this period the patients are contagious. Only after the resolution of the rash and the disappearance of each lesion of the skin, a person can be considered cured.

Atypical features have also been reported [85]:-patients without prodromal period (appearance of rash before the onset of signs and symptoms)-patients with a modest size rash-patients without rash but with anal pain and bleeding-patients with rash present only in the genital or perineal/perianal skin area-patients with rash at different stages of development

Infrequently, complications such as encephalitis, pneumonitis, digestive involvement, ocular diseases leading to loss of visual acuity, kidney injury, skin and soft tissue infections and abscesses may occur over the first 5–28 days of illness (Figure 1) [50,86,87]. Another rare occurrence is the myopericarditis directly due to the virus or through immune-mediated injury [88].

Babies, children, individuals with underlying immune deficiencies and pregnant women may develop severe disease [70]. There are different scientific opinions about an increase of the risk of infective complications in patients with monkeypox disease affected by HCV and HIV [89].

Death is a possible event, especially in this latter cluster of people [70,84].

## 6. Diagnosis

The diagnosis may be difficult since it may be hard to distinguish Monkeypox from other exanthematous viruses, particularly varicella and variola.

The diagnosis of monkeypox takes advantage of clinical and laboratory findings (Figure 1). The clinical phase is based on the presence of lymphadenomegaly, pre-rash fever, myalgia, headache, back pain, slower maturation of skin lesions, whereas the laboratory phase is focused on virus detection.

The main laboratory test includes the execution of nucleic acid amplification testing, with real time polymerase chain reaction (PCR), of the exudate of the skin lesions, pharyngeal swabs and seminal fluid of the patient [83,90]. According to common guidelines, this procedure should be performed in two swabs from at least three lesions. In doing so, PCR is accredited of high sensitivity and specificity but, unfortunately, this test is not yet available everywhere [84].

Some laboratory findings are of little help. Although, frequently, hypoalbuminemia, high transaminase levels, low values of blood urea nitrogen can be found [78].

Tests for antigens or antibodies research may also be used to diagnose monkeypox but, due to crossreactivity with other orthopoxviruses, they have low sensitivity and specificity [83]. However, in the case of orthopoxvirus-induced encephalitis, they can be very helpful to evaluate the anti-orthopoxvirus IgM reaction in cerebrospinal fluid and the subsequent seroconversion to the anti-orthopoxvirus IgG response [86].

The analysis of cerebrospinal fluid initially may prevalently show a polymorphonuclear pleocytosis, with normal glucose and protein levels. However, within 5–6 days, there may be a change as pleocytosis can become mainly of lymphocytic type [86,91].

Cerebral Magnetic Resonance imaging may show diffuse edema, meningeal enhancement, and increased signal in the thalamus and parietal cortex on fluid-attenuated inversion recovery sequences, whereas slow waves can be observed on electroencephalogram [91].

Histology is unhelpful due to overlapping histologic features with those of viruses such as herpes (especially herpes simplex and varicella), vaccinia and cowpox.

Instead, immunohistochemistry permits to distinguish herpes from pox virus. Indeed, monoclonal antibodies are available that may detect orthopox antigen and monkeypox virus [92].

## 7. Management and Strategies

No specific Food and Drug Administration (FDA) approved treatment exists for monkeypox disease, but some recommendations should be followed, especially with the aim to prevent or better manage the development of possible complications [84].

The symptomatic management of fever and respiratory distress, with antipiretics and supplemental oxygen, is desirable where necessary [86].

Encephalitis accompanying monkeypox, depending on symptoms and causes, needs to be treated with intravenous fluids, supplemental oxygen, steroids, antiseizure medicine, antiviral or antibiotics.

Secondary bacterial infections (pneumonia, abscess, other infections of the skin) should be treated with empirical oral or parenteral cephalosporins or beta-lactam antibiotics [70].

No FDA approved antivirals exist to treat monkeypox, although tecovirimat is FDA approved for treating smallpox, being able to prevent viral release from cells via inhibition of the viral envelope protein p37. However, even if current studies have not proven the effectiveness of tecovirimat in human monkeypox, a recent report supports the continued recourse to treatment with tecovirimat in the current monkeypox outbreak, also taking into account that it is generally well tolerated [93]. Moreover, other authors suggest that it may be useful in patients with weakened immune systems without age limits [34,78]. Finally, CDC guidelines advise its use in patients with severe disease, or in those at high risk for severe disease, or with aberrant infections [46]. However, headache, nausea, abdominal pain, vomiting, neutropenia and infusion site reaction are some side effects that patients might experience after the antiviral administration [78].

Although the use of monoclonal antibodies is approved to treat severe cases of monkeypox and subjects affected by T cell lymphopenia, divergent views exist on their safety. Indeed, they are burdened with some problems in the production methods, including the one of being exposed to the risk of contamination with blood-borne infectious agents [78,94].

## 8. Principal Differences with COVID-19 Pandemic

COVID-19 epidemic is one of those unexpected and inscrutable events hard to manage since we had to deal with the appearance of a completely new virus capable of causing the death of over 15 million people worldwide [93].

Some features have made COVID-19 very insidious, such as [95,96,97]:-being an RNA virus, the mechanisms of mutation repair are less effective and allow the formation of new more contagious strains more easily-the easy dissemination to animals in which new and more dangerous mutations may occur that make the COVID-19 even more difficult to combat once it is reintroduced in mankind-the possibility that it can be transmitted to others during the incubation period of the infected subject-the airborne transmission through aerosol inhalation-the ease of contagion between people-after healing from the acute phase, the prolongation of the disease in a chronic phase (the so called long COVID).-the possible increased risk of developing over the years other diseases (psychiatric illnesses, diabetes, heart diseases, stroke).

The availability of vaccines and drugs, as well as the acquired immunity among people allowed to reduce the number of deaths to more acceptable levels. However, uncertainty remains on the possible new appearance of a COVID strain resistant to human immune system.

At present, this makes it difficult to see the definitive outcome of this pandemic phase until a new vaccine can provide protection against all potential COVID strains.

## 9. Conclusions

Human monkeypox, previously classified as a rare zoonotic disease, raised awareness within the worldwide public health because of its rapid growth in non-endemic nations in which it may possibly create life threatening situations [71,98] depending on the fact that 70% of the global population is no longer vaccinated against smallpox. This large portion of susceptible population has favored the current monkeypox outbreak [90].

According to some researchers, during the current outbreak, the isolated clade of monkeypox could cause in the majority of cases a mild self-limiting disease, whereas only some of them were hospitalized, mainly for pain [14]. However, the prevalent opinions agree that monkeypox is a serious public health threat either because of the possibility of bioterrorism or due to the occurrence of natural outbreaks that may particularly affect high-risk subjects.

The lack of resources, diagnostic equipment (adequate laboratory infrastructure), procedures (weak disease surveillance systems) and adequate knowledge of this disease in African countries, with the result of not having uniform health controls of people, (as it was in the most developed countries) was also crucial to favor the worldwide transmission through the international traffic [99].

A few months ago, the WHO expressed serious concern regarding the epidemiological situation in Europe, classifying it to be at high risk due to the spreading of the monkeypox in 30 European countries, where it infected more than 6000 subjects [100]. From January to October 2022, the global situations of monkeypox cases and deaths in non-endemic countries is well described in Table 2 [101].

However, the latest reports of the WHO show that the total number of monkeypox cases reported in the European regions and in the Americas it is decreasing, with the American regions that continue to hold the highest cumulative proportion of monkeypox cases reported globally [101].

In order to better contain the expansion of the monkeypox outbreak preventing it from continuing to spread throughout the world, early recognition planning screening protocols adapted to local settings, prompt that isolation and immediate hygiene measures should be adopted without hesitation in endemic countries.

Moreover, the vaccination should be recommended as it is considered to be an effective countermeasure to break the chain of transmission and/or to prevent the serious form of the disease [102].

Indeed, previous trials demonstrated that smallpox vaccine is highly effective in preventing monkeypox disease, not losing efficacy also after a few years by protecting subjects against the occurrence of a severe illness [99,103].

It is of fundamental importance to implement the investments in public health of endemic African countries and launch information campaign to promote appropriate health behaviors that may help stop the disease.

To this purpose, it is crucial to prioritize high-risk populations, such as communities of men who have sexual contact with a new or multiple male partners. Due to their intense relations also through sexual networks, these communities represent the most affected category in this epidemic in non-endemic countries [104,105]. Therefore, in these countries, it is appropriate that all the necessary assistance is provided assuring these communities have effective information and services, as well as taking measures to protect their health and dignity. At the same time, the challenge is also to put in place all precautions designed to ensure the protection of the public health, both defining what can be clinically effective against this disease [106] and making proper use of current health knowledge on the subject.

## Figures and Tables

**Figure 1 ijerph-19-16781-f001:**
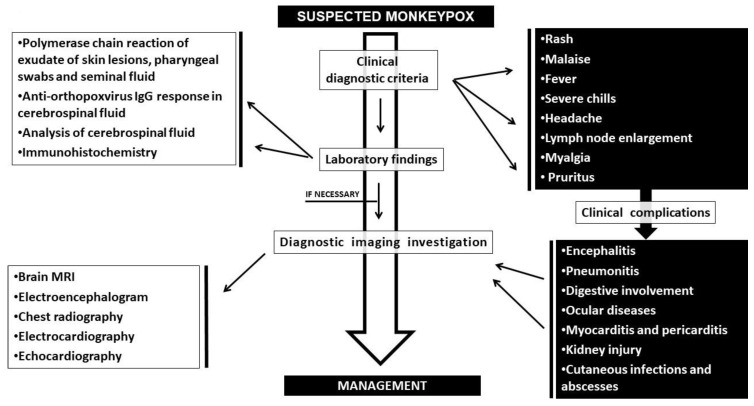
A diagram of clinical signs, diagnostic methods and potential complications of monkeypox disease.

**Table 1 ijerph-19-16781-t001:** The number of cumulative confirmed monkeypox cases reported to the WHO, in non-endemic Regions, from 1 January 2022 to July 2022.

WHO	NON-ENDEMIC COUNTRIES	CASES REPORTED
EUROPEAN REGION	Austria	12
	Belgium	77
	Czechia	6
	Denmark	13
	Finland	4
	France	277
	Georgia	1
	Germany	521
	Gibraltar	1
	Greece	3
	Hungary	7
	Iceland	3
	Ireland	24
	Italy	85
	Latvia	2
	Luxembourg	1
	Malta	2
	Netherlands	167
	Norway	4
	Poland	7
	Portugal	317
	Romania	5
	Serbia	1
	Slovenia	8
	Spain	520
	Sweden	13
	Switzerland	46
	The United Kingdom	793
MAINLAND OF THE AMERICAS	Argentina	3
	Brazil	11
	Canada	210
	Chile	3
	Mexico	11
	United States of America	142
	Venezuela	1
MEDITERRANEAN REGION	Israel	13
	Lebanon	1
	Marocco	1
	United Arab Emirates	13
ASIA	Australia	9
	Republic of Korea	1
	Singapore	1
REPUBLIC OF CHINA	Taiwan	1

**Table 2 ijerph-19-16781-t002:** The number of cumulative confirmed monkeypox cases and deaths reported to the WHO, by the WHO from 1 January 2022 to 30 October 2022.

WHO	DEATHS	CASES REPORTED
EUROPEAN REGION	4	25.303
EASTERN MEDITERRANEAN REGION	1	72
MAINLAND OF THE AMERICAS	16	50.176
SOUTH EAST ASIA REGION	1	30
WESTERN PACIFIC REGION	0	209

## Data Availability

See References list.

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
