# Peer review of "Human Monkeypox—A Global Public Health Emergency"

_ijerph, 2022, doi:10.3390/ijerph192416781_

Round 1
Reviewer 1 Report
Dear Authors:
Attached please find the reviewed version (pdf) of the MS. There you can find all the comments, suggestions, and questions I had regarding your MS. Try your best to revise your MS accordingly.
Best Wishes

Author Response
Reviewer 1
1. need to set proper spacing between lines
- Corrected
- []
- Corrected
- ACAM 2000: have you defined this term before abbreviating it here
- It is the trade name and therefore it is not possible to define it.
4.MVA-BN
- It is the trade name and therefore it is not possible to define it.
- LC16m8
- It is the trade name and therefore it is not possible to define it.
- spacing issue
- Corrected
- Change in Font size is observed
- Modified the text according to the general format
- Looks like the hyperlinks were not removed. This indicates if there is copy paste do from some source. most probably on the web
- Removed the hyperlinks and modified the text according to the general format.
- need to set proper spacing between lines
- Corrected
- Font is in bold
- Removed the bold
- spacing issue
- Corrected
- spacing issue
- Corrected
- spacing issue
- Corrected
- The hyperlink was not removed
- Removed the hyperlink and corrected the text according to the general format.
- Change in Font size is observed
- Modified the text according to the general format
- Here and elsewhere the journal abbreviations should be typed in italic font
- Corrected
- You can use et al. after the 8th author in such types of articles
- Done
Reviewer 2 Report
This review paper is good, but I have some suggestions, Please read the paper line by line and correct English grammar mistakes.
Add more keywords
Please explain the mechanism of virus-host infection and add at least one mechanistic diagram also, presenting the potential targets of monkeypox disease.
Add one paragraph by Keeping your focus on how this infection is different from COVID-19 and how many serious outcomes could be tolerated and explain the maximum strategies that could follow to control this disease worldwide.
Add a paragraph about the proposed drugs/vaccines for Monkeypox infections.
Please read the published data improve your paper, and also consider these relevant papers:
doi: 10.1016/j.ijsu.2022.106896
doi: 10.3389/fimmu.2022.1035924
Thank you.
Author Response
Reviewer 2
This review paper is good, but I have some suggestions, Please read the paper line by line and correct English grammar mistakes.
- Done
Add more keywords
- I added other 6 keywords
Please explain the mechanism of virus-host infection and add at least one mechanistic diagram also, presenting the potential targets of monkeypox disease.
- I agree with the reviewer and added a phrase in the first lines of “Pathophysiology and Clinical Features” section, that specifies better the mechanisms of virus-host infection already mentioned. Also I added a diagram ( figure 1).
Add a paragraph about the proposed drugs/vaccines for Monkeypox infections.
- The subject “vaccines” was already covered within the paragraph 3 “Prevention”. However, according to your suggestions I put this argument in a new paragraph 4 “Proposed vaccines”and added some phrases.
Please read the published data improve your paper, and also consider these relevant papers:
doi: 10.1016/j.ijsu.2022.106896
doi: 10.3389/f.immu.2022.1035924
In accordance with your suggestions I quoted both articles.
Reviewer 3 Report
1. Is there a difference between European countries in relation to the degree of complete vaccination against smallpox?
2. How is such a high prevalence of the disease in the United Arab Emirates connected in comparison with countries that have a significantly larger gay population?
3. Did the HIV positive patients have detectable At or were they PCR negative, and were they all previously treated?
4. It is noticeable that in certain regions of the world mortality is higher compared to some parts. Did the death occur as a result of non-treatment or some co-morbidity?
Author Response
Reviewer 3
- Is there a difference between European countries in relation to the degree of complete vaccination against smallpox?
In accordance with your interesting suggestions I added a sentence that answers this question in paragraph 4.
- How is such a high prevalence of the disease in the United Arab Emirates connected in comparison with countries that have a significantly larger gay population?
As you can see in table 1 the cases reported in United Arab Emirates are very few compared to those reported in European countries.
- Did the HIV positive patients have detectable At or were they PCR negative, and were they all previously treated?
It is an interesting question but in the works viewed this detail is not clearly specified and I can not answer.
4.It is noticeable that in certain regions of the world mortality is higher compared to some parts. Did the death occur as a result of non-treatment or some co-morbidity?
In a recent report that I quoted to the end of paragraph 2 the most of patients that died in USA were treated.
Round 2
Reviewer 1 Report
No more comments
Reviewer 3 Report
Dear colleague,
I think that you have not answered all the suggestions that I have given you that would be of importance to the readers
Kind regards